# Leveraging AutoML for Sustainable Deep Learning: A Multi-Objective HPO Approach on Deep Shift Neural Networks

## Abstract

Deep Learning (DL) has advanced various fields by extracting complex patterns from large datasets. However, the computational demands of DL models pose environmental and resource challenges. Deep Shift Neural Networks (DSNNs) improve the situation by leveraging shift operations to reduce computational complexity at inference. Compared to common DNNs, DSNNs are still less well understood and less well optimized. By leveraging AutoML techniques, we provide valuable insights into the potential of DSNNs and how to design them in a better way. Since we consider complementary objectives such as accuracy and energy consumption, we combine state-of-the-art multi-fidelity (MF) HPO with multi-objective optimization to find a set of Pareto optimal trade-offs on how to design DSNNs. Our approach led to significantly better configurations of DSNNs regarding loss and emissions compared to default DSNNs. This includes simultaneously increasing performance by about 20% and reducing emissions by about 10%. Investigating the behavior of quantized networks in terms of both emissions and accuracy, our experiments reveal surprising model-specific trade-offs, yielding the greatest energy savings. For example, in contrast to common expectations, selectively quantizing smaller portions of the network with low precision is optimal while retaining or improving performance. We corroborated these findings across multiple backbone architectures, highlighting important nuances in quantization strategies and offering an automated approach to balancing energy efficiency and model performance.

## 1 Introduction

Deep Learning (DL) is a promising approach to extracting information from large datasets with complex structures. This includes performing computations in IoT environments and on edge devices (Li et al., 2018; Zhou et al., 2019), which can come with strict limitations on the energy consumption. However, with the ever-increasing size and performance of such models due to the progress in science and industry, running these models is not free of computational costs (Sze et al., 2017) and minimizing this cost directly affects a model's environmental impact (Schwartz et al., 2020). Even if resource consumption should not be considered crucial in view of environmental impact, efficiently designed neural networks free up resources that can be used for other tasks, e.g., edge computing or computations for advanced driver assistance systems (Howard et al., 2017). With our approach, we contribute to DL by minimizing its environmental footprint and allow applications in low-resource settings. Of particular interest to us are Deep Shift Neural Networks (DSNNs) that offer great potential in reducing power consumption compared to traditional Deep Learning models, e.g., by reducing the inference time by a factor of 4 (Elhoushi et al., 2021). Instead of expensive floating point arithmetic, they leverage cheap shift operations — specifically, bit shifting — as the computational unit, which boosts efficiency by replacing costly multiplication operations in convolutional networks. Although DSNNs offer great promise, so far, the different design decisions, including training hyperparameters and shift architecture, have not been well-studied and there is little knowledge about their full potential. We suspect that the configuration of DSNNs has a huge impact on both performance and computational efficiency.

One of the key challenges with DSNNs is determining the appropriate level of precision for shift operations to minimize quantization errors without excessively increasing the computational load.

To address this challenge, we propose to apply automated machine learning (AutoML) to DSNNs to find their optimal configuration. This is achieved by hyperparameter optimization (HPO) (Bischl et al., 2023) and a neural architecture search on a macro-level (Elsken et al., 2019). Integrating multi-fidelity (MF) and multi-objective optimization (MO) techniques (Belakaria et al., 2020) facilitates an optimal exploration of the configuration space that trades off predictive performance and energy consumption (Deb, 2014). To this end, we extended the SMAC3 approach (Lindauer et al., 2022), a state-of-the-art HPO package (Eggensperger et al., 2021), such that its MO implementation effectively balances the trade-off between achieving high predictive accuracy and minimizing energy consumption. Employing tools like CodeCarbon (Lacoste et al., 2019; Lottick et al., 2019) during the training and evaluation phases provides insights into the energy consumption and carbon emissions associated with each model configuration. The MF aspect allows for the efficient use of computational resources by evaluating configurations at varying levels of approximation. Our work is in the spirit of Green AutoML (Tornede et al., 2023) by considering efficient AutoML for gaining insights into the design of efficient DSNNs.

**Contributions.** Overall, we contribute to Green AutoML w.r.t. DSNNs by:

1. Specifying the first configuration space tailored to DSNNs which is efficiently optimized by a combination of multi-objective and multi-fidelity AutoML approaches;

2. Empirically exploring how specific design choices in DSNNs lead to different trade-offs between accuracy and energy efficiency, enabling stakeholders and researchers to leverage these findings to develop energy-efficient applications that maintain high computational accuracy; and

3. Identifying specific configurations of DSNNs that surpass the baseline results in both dimensions of the performance-efficiency optimization problem.

## 2  RELATED WORK

Both multi-fidelity optimization (MF) (Bischl et al., 2023) and multi-objective optimization (MO) (Morales-Hernández et al., 2022) for AutoML have gotten a lot of traction in recent years. The combined integration of multi-fidelity multi-objective optimization (MFMO) has seen some advancements to enhance the efficiency of model training while minimizing environmental impact. Belakaria et al. (2020) proposed an acquisition function based on output space entropy search for multi-fidelity multi-objective Bayesian optimization (MFMO-BO-OSES). Their method addresses the exploration-exploitation dilemma by prioritizing the acquisition of data points that significantly reduce the entropy of the Pareto front. This approach enables more strategic sampling decisions and leverages lower-fidelity evaluations to approximate the Pareto front effectively, aligning with the sustainability goals of Green AutoML. Similarly, Schmucker et al. (2020) considered a combination of multi-objective and multi-fidelity optimization but focused on fairness as the second objective. With our MFMO approach, we contribute an algorithm tailored specifically for performing efficient HPO tasks using BO, directly minimizing emissions in the process.

A well-explored technique for reducing the computational complexity of neural networks themselves is network quantization. It involves lowering the precision of weights and activations, which decreases the model's memory footprint and accelerates inference. Works by Courbariaux et al. (2015) and Rastegari et al. (2016), for example, have demonstrated that techniques such as BinaryConnect and XNOR-Net not only reduce computational requirements but also maintain near state-of-the-art performance, underscoring the potential of quantization to balance performance with computational efficiency. This has also been transferred into the field of LLMs, where 1-Bit transformer architectures are used to address the challenges of increasing model size (Wang et al., 2023). Strongly related to network quantization, Deep Shift Neural Networks (DSNNs), proposed by Elhoushi et al. (2021), represent an advancement towards quantization combined with efficient operators. DSNNs employ bitwise shift operations instead of traditional multiplications, thus further reducing the computational overhead and power consumption. This innovation is particularly crucial for deploying Deep Learning models in power-sensitive or resource-constrained environments, further contributing to the environmental sustainability of AI technologies.

DSNNs offer a more balanced trade-off between computational efficiency and accuracy compared to binary methods like BinaryConnect and XNOR-Net by replacing multiplications with bitwise

shifts, which are more power-efficient yet maintain greater precision, reducing the accuracy loss typically associated with binary quantization. This allows DSNNs to achieve better performance in resource-constrained environments while still minimizing computational overhead (Elhoushi et al., 2021).

AdderNet (Chen et al., 2020) is another approach for reducing the amount of computationally expensive multiplications during network training and inference by using the $\ell_1$-norm distance between input and filter vectors to compute activations. There are efforts in mixed-precision quantization, where different bit-widths are assigned to different layers or channels of the network. It allows for higher precision where necessary and lower precision where possible, optimizing the trade-off between accuracy and efficiency (Motetti et al., 2024). Similarly, ternary quantization constrains the weights to three discrete values: $\{-1, 0, +1\}$. To mitigate the accuracy loss from reduced precision, approaches like trained ternary quantization (TTQ) introduce learned scaling factors, allowing networks to maintain performance comparable to their full-precision counterparts while significantly reducing memory and power usage (Rokh et al., 2023). HPO combined with DSNNs further optimizes both performance and resource efficiency by tailoring shift depths and quantization strategies, allowing for fine-tuned control over energy consumption and accuracy in constrained environments, making them ideal for mixed-precision and quantization-aware neural network applications.

## 3 BACKGROUND

The following chapter introduces foundational concepts for our approach.

### 3.1 DEEP SHIFT NEURAL NETWORKS

A Deep Shift Neural Network (DSNNs) is an approach to reduce the computational and energy demands of Deep Learning (Elhoushi et al., 2021). They achieve a considerable reduction in latency time by simplifying the network architecture such that they replace the traditional multiplication operations in neural networks by bit-wise shift operations and sign flipping, making DSNNs suitable for computing devices with limited resources. There are two methods for training DSNNs (Elhoushi et al., 2021): DeepShift-Q (Quantization) and DeepShift-PS (Powers of two and Sign). DeepShift-Q involves training regular weights constrained to powers of 2 by quantizing weights to their nearest power of two during both forward and backward passes. In DeepShift-Q, the weights are quantized to powers of two by rounding the logarithm of the absolute weights to the nearest integer. This process simplifies the weight representation and ensures compatibility with bitwise shift operations. The sign is then applied to preserve the original weight polarity. DeepShift-PS directly includes the values of the shifts and sign flips as trainable hyperparameters, offering finer control over weight adaptation. This approach removes the need for explicit rounding during training, potentially leading to improved precision at the cost of additional parameter updates.

The DeepShift-Q approach obtains the sign matrix $S$ from the trained weight matrix $W$ as $S = sign(W)$. The power matrix $P$ is the base-2 logarithm of $W$'s absolute values, i.e., $P = \log_2(|W|)$. After rounding $P$ to the nearest power of two, $P_r = round(P)$, the quantized weights $\tilde{W}_q$ are calculated by applying the sign from $S$, shown as

$$\tilde{W}_q = flip(2^{P_r}, S) . \tag{3.1}$$

The DeepShift-PS approach optimizes neural network weights by directly adapting the shift ($\tilde{P}$) and sign ($\tilde{S}$) values. The shift matrix $\tilde{P}$ is obtained by rounding the base-2 logarithm of the weight values, $\tilde{P} = round(P)$, and the sign flip $\tilde{S}$ is computed as $\tilde{S} = sign(round(S))$. Weights are calculated as

$$\tilde{W}_{ps} = flip(2^{\tilde{P}}, \tilde{S}) , \tag{3.2}$$

where the sign flip operation $\tilde{S}$ assigns values of $-1$, $0$, or $+1$ based on $s$.

Directly training shift and sign values could allow for more precise control in optimizing the network's computational efficiency by reducing mathematical imprecision. On the other hand, training the floating point weights and only rounding them during the forward and backward pass might increase the precision and reduce the error in training the weights.

## 3.2 HYPERPARAMETER OPTIMIZATION

The increasing complexity of Deep Learning algorithms enhances the need for automated hyperparameter optimization (HPO) to increase model performance (Bischl et al., 2023). Consider a dataset $\mathcal{D} = \{(x_i, y_i)\}_{i=1}^{N} \in \mathbb{D} \subset \mathcal{X} \times \mathcal{Y}$, where $\mathcal{X}$ is the instance space and $\mathcal{Y}$ is the target space, and a hyperparameter configuration space $\Lambda = \{\lambda_1, \ldots, \lambda_L\}$, $L \in \mathbb{N}$. In our work, $\mathcal{M}$ denotes the space of possible DSNN models. The dataset $\mathcal{D}$ is split into disjunct training, validation, and test sets: $\mathcal{D}_{train}, \mathcal{D}_{val}$, and $\mathcal{D}_{test}$ respectively. An algorithm $\mathcal{A} : \mathbb{D} \times \Lambda \to \mathcal{M}$ trains a model $M \in \mathcal{M}$, instantiated with a configuration of $L$ hyperparameters sampled from $\Lambda$, on the training data $\mathcal{D}_{train}$. The performance of the algorithm is assessed via an expensive-to-evaluate loss function $\mathcal{L} : \mathcal{M}_\lambda \times \mathbb{D} \to \mathbb{R}$, which involves both the training on $\mathcal{D}_{train}$ and the evaluation of the model on $\mathcal{D}_{val}$. The direct optimization objective of HPO is to find the configuration $\lambda^* \in \Lambda$ with minimal validation loss $\mathcal{L}$, such that:

$$\lambda^* \in \arg\min_{\lambda \in \Lambda} \mathcal{L}\big(\mathcal{A}(\mathcal{D}_{train}, \lambda), \mathcal{D}_{val}\big). \tag{3.3}$$

Finally, the model's final performance is assessed on $\mathcal{D}_{test}$.

## 3.3 BAYESIAN OPTIMIZATION

For a given dataset, Bayesian Optimization (BO) for HPO is a strategy for global optimization of black-box loss functions $\mathcal{L}(\lambda) : \mathcal{M}_\lambda \times \mathbb{D} \to \mathbb{R}$ that are expensive to evaluate (Jones et al., 1998).

BO uses a probabilistic surrogate model $\mathcal{S}$ to approximate the loss function, commonly given by a Gaussian Process or a Random Forest (Rasmussen & Williams, 2006; Hutter et al., 2011; Shahriari et al., 2016). An acquisition function $\alpha : \Lambda \to \mathbb{R}$ guides the search for the next optimal evaluation points by balancing the exploration-exploitation trade-off, based on the set of previously evaluated configurations $\{(\lambda_1, \mathcal{L}_1), ..., (\lambda_{m-1}, \mathcal{L}_{m-1})\}$ at time $m$. Common choices for acquisition functions include expected improvement (EI) (Jones et al., 1998) since it calculates the expected improvement in the objective function value and guides the search towards regions where improvements are most likely.

Entropy-based methods like Entropy Search (ES) (Hennig & Schuler, 2012) and Predictive Entropy Search (PES) (Hernández-Lobato et al., 2014) aim to reduce the entropy of the posterior distribution of the maximizer, focusing on information-rich regions.

The Knowledge Gradient (KG) (Frazier et al., 2009) offers a strategy for maximizing the expected improvement of the objective considering all potential outcomes, valuable in scenarios with noisy measurements. BO is particularly well-suited for hyperparameter optimization in Deep Learning, where evaluating the performance of a model configuration can be computationally expensive because of the training of each configuration. BO is sample-efficient in evaluating $\mathcal{L}$ on only a few configurations.

## 3.4 MULTI-FIDELITY OPTIMIZATION

Since it is not feasible to fully train multiple configurations of DSNNs for comparison due to computational efficiency, we employ a multi-fidelity (MF) approach (Li et al., 2017), which is a common strategy in AutoML to navigate the trade-off between performance and approximation error (Hutter et al., 2019). MF approaches train cheap-to-evaluate proxies of black-box functions following different heuristics, e.g., allocating a small number of epochs to many configurations in the beginning and training the best-performing ones on an increasing number of epochs. Formally, we define a space of fidelities $\mathcal{F}$ and aim to minimize a function $F \in \mathcal{F}$ (Kandasamy et al., 2019):

$$\min_{\lambda \in \Lambda} F(\lambda). \tag{3.4}$$

We approximate $F \in \mathcal{F}$, using a series of lower-fidelity, i.e., less expensive approximations $\{f(\lambda)_1, \ldots, f(\lambda)_j = F(\lambda)\}$, where $j$ denotes the total number of fidelity levels. The target function $F \in \mathcal{F}$ corresponds to the loss function $\mathcal{L}$ of HPO and BO. The allocated resources for evaluating a model's performance at various fidelities are referred to as a budget, e.g., training a DNN for only $n$ epochs instead of until convergence. MF typically assumes that the highest fidelities approximate the black-box function best. The longer a model is trained, the more accurate its approximation of an underlying function gets.

## 3.5 MULTI-OBJECTIVE OPTIMIZATION

Multi-objective optimization (MO) addresses problems involving multiple, often competing, objectives. This approach is used in scenarios where trade-offs between two or more conflicting objectives must be navigated, such as, in the context of DSNNs, enhancing accuracy alongside reducing energy consumption. MO aims to identify Pareto optimal solutions (Deb, 2014). New points are added based on the current observation dataset $\mathcal{D}_{obs} = \{(\lambda_1, \mathcal{L}(\lambda_1)), \ldots, (\lambda_n, \mathcal{L}(\lambda_n))\}$ at time $n + 1$. These points augment the surface formed by the non-dominated solution set $D_n^\star$, which satisfies the condition for $d$ objective variables and a loss function $\mathcal{L} = (\mathcal{L}_1, \ldots, \mathcal{L}_d)$, where $\mathcal{L}_k$ corresponds to the loss regarding objective $k$ (Wada & Hino, 2019):

$$\forall \lambda, (\lambda, \mathcal{L}(\lambda)) \in \mathcal{D}_n^\star \subset \mathcal{D}_n, \ (\lambda', \mathcal{L}(\lambda')) \in \mathcal{D}_n \tag{3.5}$$
$$\exists k \in \{1, \ldots, d\} : \ \mathcal{L}_k(\lambda) \leq \mathcal{L}_k(\lambda').$$

W.l.o.g. we assume the minimization of all objectives. The observation dataset $\mathcal{D}_{obs}$ is iteratively updated to search for solutions that approximate the Pareto front.

## 4 APPROACH

Our goal is to provide insights into the structure of DSNNs. We want to optimize these for performance and efficiency and show how their specific hyperparameters affect that optimization.

### 4.1 CONFIGURATION SPACE EXPLORATION

The foundation of our approach lies in defining and exploring a robust configuration space tailored specifically for DSNNs. This space includes a range of hyperparameters that influence the network's performance and energy efficiency. Key hyperparameters under consideration include:

**Shift Depth** determines the number of network layers converted to employ shift operations, replacing conventional floating point operations and thereby reducing computational overhead.

**Shift Type** selects the method of shift operation, either quantization (DeepShift-Q) or direct training of shifts (DeepShift-PS), impacting the network's training dynamics and inference efficiency.

**Bit Precision for Weights and Activations** influences the network's accuracy and the granularity of its computations, affecting both performance and power consumption.

**Rounding Type** affects how weight adjustments are handled during training, with options for deterministic or stochastic rounding, each offering trade-offs in terms of computational stability and performance.

Table 1 details the configuration space for a ResNet20 model adapted for DSNNs, outlining the range and default values of each hyperparameter considered in our study.

### 4.2 MULTI-FIDELITY MULTI-OBJECTIVE OPTIMIZATION FRAMEWORK

To computationally enhance Deep Shift Neural Networks (DSNN) via AutoML, we employ multi-fidelity optimization (MF), see Section 3. A well-known MF algorithm is successive halving (Jamieson & Talwalkar, 2016), where $n_c$ configurations are trained on an initial small budget $b_I$. It addresses the trade-off between $b_I$ and $n_c$, or between approximation error and exploration inherent in successive halving, using the HyperBand algorithm for MF. HyperBand (Li et al., 2017) runs successive halving in multiple brackets, where each bracket provides a combination of $n_c$ and a fraction of the total budget per configuration so that they sum up to the total budget.

We extend this to multi-fidelity multi-objective optimization (MFMO). We simultaneously address the accuracy of the model as well as its energy consumption using a two-dimensional objective function:

$$\mathcal{L}_{\text{MO}} : \Lambda \to \mathbb{R}^2, \quad \mathcal{L}_{\text{MO}}(\lambda) = \big(\mathcal{L}_{\text{loss}}(\lambda), \mathcal{L}_{\text{emission}}(\lambda)\big), \tag{4.1}$$

where, given a configuration $\lambda \in \Lambda$, $\mathcal{L}_{\text{loss}}(\lambda)$ aims to minimize the loss, enhancing the model's accuracy, and $\mathcal{L}_{\text{emission}}(\lambda)$ seeks to minimize the energy consumption during training and inference, promoting environmental sustainability. We aim to solve the following optimization problem:

$$\arg \min_{\lambda \in \Lambda} \mathcal{L}_{\text{MO}}(\lambda). \tag{4.2}$$

Our approach utilizes the MFMO framework to efficiently navigate the defined configuration space with less computationally expensive proxies of the full training regimen—that enable a broader exploration of the hyperparameter space within feasible computational limits.

### 4.3 ALGORITHMIC IMPLEMENTATION OF MFMO

We use the ParEGO algorithm (Knowles, 2006) to compute Pareto optimal configurations. It transforms the multi-objective problem into a series of single-objective problems by introducing varying weight hyperparameters for the objectives in each iteration of HyperBand. Thus optimizing a different scalarization per evaluation to approximate the Pareto front. The resulting single-objective optimization function can then be evaluated in an MF setting. All configurations having survived a successive halving bracket are checked against the current Pareto front approximation and the Pareto set is updated if necessary. The computational strategy initially involves computing a broad array of configurations and leveraging the successive halving method to efficiently narrow down the field to the most promising candidates. We specifically target solutions that represent both extremes of the Pareto front—those that excel in one objective at the potential expense of the other—and configurations that provide a balanced compromise between the two objectives. For easier understanding of our approach, we included pseudocode of our algorithmic implementation in Algorithm 1 in the appendix.

Table 1: Configuration search space of ResNet20. The first half includes commonly used training hyperparameters for DL, whereas the second half is specific to DSNNs.

| Hyperparameter | Search Space | Default |
|---|---|---|
| Batch Size | [32, 128] | 128 |
| Optimizer | {SGD, Adam, Adagrad, Adadelta, RMSProp, RAdam, Ranger} | SGD |
| Learning Rate | [0.001, 0.1] | 0.1 |
| Momentum | [0.0, 0.9] | 0.9 |
| Weight Decay | [1e-6, 1e-2] | 0.0001 |
| Weight Bits | [2, 8] | 5 |
| Activation Integer Bits | [2, 32] | 16 |
| Activation Fraction Bits | [2, 32] | 16 |
| Shift Depth | [0, 20] | 20 |
| Shift Type | {Q, PS} | PS |
| Rounding | {deterministic, stochastic} | deterministic |

## 5 EXPERIMENTS

In the following section, we detail the setup and methodology used to evaluate our approach discussed in Section 4, focusing on optimizing Deep Shift Neural Networks (DSNNs) through multi-fidelity, multi-objective optimization (MFMO), and extending the DSNN objective function to multi-objective to compute a Pareto front for optimality regarding performance and efficiency. We discuss how our approach successfully navigates the model performance and environmental impact trade-offs. From this, we gain insights into DSNNs and how specific design choices might affect their performance. By identifying optimal configurations for both or either objectives, we draw conclusions about how the DSNN specific hyperparameters in the network architecture interact with each other.

### 5.1 EVALUATION SETUP

We train and evaluate our models on the CIFAR10 dataset (Krizhevsky et al., 2009) and the Caltech101 dataset (Fei-Fei et al., 2004), using NVIDIA A100 GPUs. For hyperparameter optimization (HPO), we extend SMAC3 (Hutter et al., 2011; Lindauer et al., 2022), as well-known state-of-the-art HPO package (Eggensperger et al., 2021). For multi-objective optimization, we aim to compute a Pareto front of optimal configurations for performance and energy consumption. To incorporate the environmental impact into our HPO workflow, we use the CodeCarbon emissions tracker (Lacoste et al., 2019; Lottick et al., 2019) to track carbon emissions from computational processes by monitoring energy use and regional energy mix in $gCO_2eq$, grams of $CO_2$ equivalents. These emission values are incorporated into SMAC3 alongside DSNN's performance metric, in this case $1-$ accuracy. As a starting point, we chose the well-known ResNet20 (He et al., 2016) architecture as

used by Elhoushi et al. (2021). Overall, this architecture is well understood and allows us to study DSNNs with few confounding factors. Additionally, we evaluate our approach using the well-known GoogLeNet (Szegedy et al., 2015) and MobileNetV2 architectures (Sandler et al., 2018). We follow the model implementation of Elhoushi et al. (2021) to ensure comparability. The configuration space is given in Table 1, for which we focus on the DSNN-specific hyperparameters (lower part of the table) and general training hyperparameters (upper part). The fidelities are the number of epochs.

## 5.2 RESULTS

### 5.2.1 QUANTITATIVE RESULTS

We first discuss the quantitive results, then the importance of the optimized hyperparameters, and finally, the implications for the configuration space. Note that we focus on the insights gained regarding DSNNs and not on how efficient our (or others') HPO approach is. In Figure 16, we present the computed Pareto fronts of a ResNet20, MobileNetVs and GoogLeNet architecture, optimized with our multi-fidelity multi-objective (MFMO) framework, on the CIFAR10 and Caltech101 datasets. Shown is a diverse set of optimal configurations that either minimize or balance the primary objectives of model accuracy and energy consumption. These are aggregated results over three seeds. The Pareto front shown results from aggregating the three individual Pareto fronts and extracting the Pareto optimal points. The default value is the mean over the loss and emissions of the default configuration on the three seeds. The Pareto fronts in Figure 16 depict how each configuration performs relative to the others within the defined hyperparameter space. The configurations were evaluated regarding classification loss and emissions emitted during inference of the model instantiated with the respective configuration.

Although we expect that Elhoushi et al. (2021) optimized their hyperparameters at least manually, we can show that our AutoML approach found even better trade-offs of the two objectives. The default configuration for the DSNN, as designed by Elhoushi et al. (2021), is in fact not part of the Pareto front in Figure 16. This holds for all architectures on both datasets. This means that there are better configurations that dominate the default configuration regarding both loss and emissions on the GoogLeNet (Figures 13 and 12), MobileNetV2 (Figures 11 and 10) and ResNet20 (Figures 15 and 14) architectures on Caltech101 and CIFAR10.

Having a closer look at Figure 15, the underlying goal of our MFMO optimization remains to balance performance and efficiency. Hence, the configurations at the bottom left of the Pareto front are especially relevant since they minimize both objectives simultaneously instead of heavily prioritizing either. There is an absolute reduction in loss of approx. 13%-33% in these configurations compared to the defaults. At the same time, relative emission reduction ranges from approx. 10%-12%. Similarly, in Figure 11, the emission reduction between the default and the next Pareto optimal solution is about 20%. Emission reductions for the other architectures range from about 5-10%. Additionally, we achieved a maximum loss reduction of about 20% for the ResNet20 on CIFAR10. This validates the need for proper HPO tuning since we found better configurations that take the energy-efficient DSNNs a significant step further by improving their accuracy and energy consumption.

### 5.2.2 HYPERPARAMETER IMPORTANCES

Another crucial aspect is the analysis of hyperparameter importance to learn their influence on a model and lay the foundation of our DSNN design insights in the next subsection. We use DeepCAVE (Sass et al., 2022) for analyzing the Pareto front of our MFMO analysis in Figure 15, and computing the hyperparameter importance using fANOVA (Hutter et al., 2014). fANOVA fits a random forest surrogate model to the hyperparameter optimization landscape and decomposes the model's variance into components corresponding to hyperparameters. This allows fANOVA to estimate the marginal impact of individual hyperparameters or pairs of hyperparameters. For an extension to MO optimization, fANOVA can be applied to each objective's performance surface separately. The hyperparameter importances are then computed for different weightings of the objectives.

Figures 8 and 9 show the hyperparameter importances of a ResNet20 on Cifar10 w.r.t. loss and emissions, respectively. The MO-fANOVA analysis for different weightings of the objectives loss and emissions can be found in Figure 17 in the appendix. The most important DSNN-specific

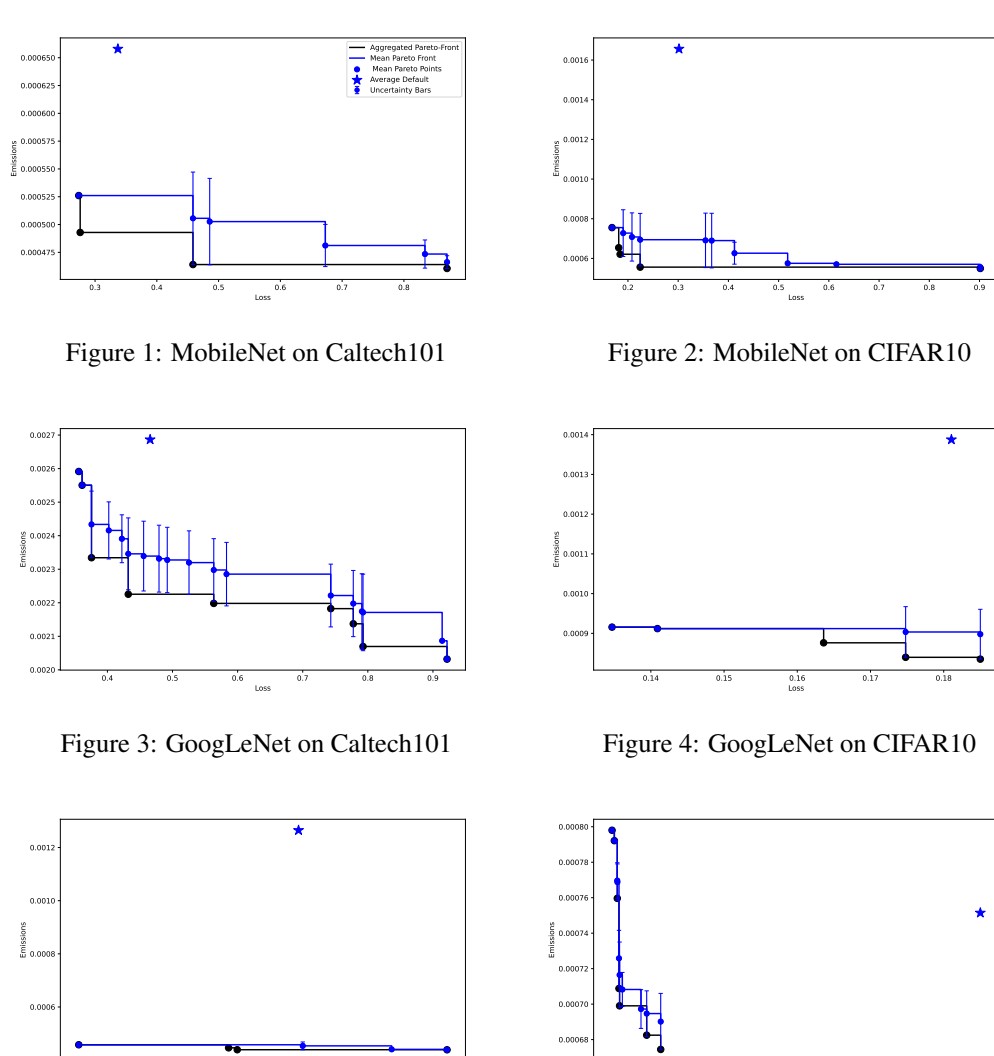

Figure 1: MobileNet on Caltech101

Figure 2: MobileNet on CIFAR10

Figure 3: GoogLeNet on Caltech101

Figure 4: GoogLeNet on CIFAR10

Figure 5: ResNet20 on Caltech101

Figure 6: ResNet20 on CIFAR10

Figure 7: Comparison of Pareto fronts for MobileNet, GoogLeNet, Resnet20 on Caltech101 and CIFAR10 datasets on multiple seed. We show the loss in % and the emissions in $gCO_2eq$. We calculate the mean Pareto front w.r.t. loss, including error bars, as well as an aggregated Pareto front of Pareto optimal solutions from all runs. The star denotes the averaged performance of the default configuration of a DSNN.

hyperparameters for emissions in Figure 9 include activation integer and fraction bits. This hints at the precision of the activation quantization being the most controlling factor for energy efficiency. Naturally, precision is a key factor since it controls the amount of operations in the network. Regarding loss in Figure 8, the shift depth is the most important hyperparameter. The proportion of the network converted to perform shift operations naturally controls the amount of information retained in the network. This is crucial for the overall performance of the network.

Notably, the shift type has a low importance value in Figures 8 and 9. This could indicate that the shift type is marginally relevant for both objectives. In practice, this insight could aid in model training by allocating fewer resources to tuning this hyperparameter, allocating resources tailored to the use case.

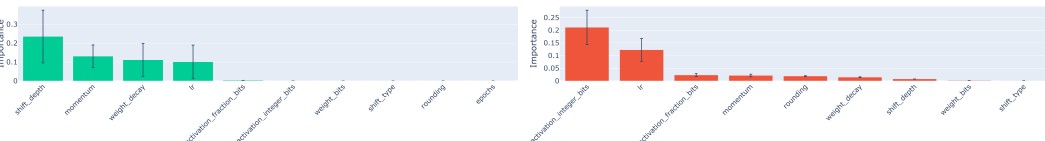

Figure 8: Hyperparameter importance according to fANOVA w.r.t. loss.

Figure 9: Hyperparameter importance according to fANOVA w.r.t. emissions.

This is supported further by looking at Tables 2 to 7. About 50% of the Pareto optimal configurations use either shift type, meaning they are not leaning toward either to maximize either objective.

With the analysis of hyperparameter importance in our study, we offer a baseline of which hyperparameters to include for future training and inference purposes. Including only the significant ones is a promising way of further boosting the energy efficiency of the DSNNs and the optimization process (Probst et al., 2019).

### 5.2.3 INSIGHTS INTO THE DESIGN OF DSNNS

Table 2 shows the specific configurations of the aggregated Pareto front of the ResNet20 architecture on CIFAR10 dataset, see the appendix for similar tables for the other architectures and datasets.

When looking at the specific configurations in Table 2, most solutions have a surprisingly small shift depth $s \in \{1, 3\}$, compared to 20 as the maximal value and the setting of the default. Hence, the Pareto optimal solutions are consistent with a very low shift depth. At the same time, the number of activation fraction bits is often rather high. This leads to the assumption that the bulk of information to be retained is in the fraction part of the activation value. A valid expectation since we used batch normalization in the ResNet20, same as Elhoushi et al. (2021). In batch normalization, the layer inputs are re-scaled and re-centered using the mean and variance of the corresponding dimension (Ioffe & Szegedy, 2015). This usually leads to small weights, highlighting the importance of activation fraction precision, which is higher in Pareto optimal configurations ranging from 8 to 32. This is still likely to be a contributing factor to the emission reduction.

Intuitively, we assumed that the shift depth is proportionate to the savings in emissions. The interaction between shift layers and other hyperparameters, such as the bit precision in weights and activations, adds another layer of complexity. However, these hyperparameters interact in a non-linear manner, influencing the model's overall energy consumption and performance in ways that are not immediately apparent. These results of the Pareto front analysis suggest that the relationship between the configuration of shift layers — a hyperparameter anticipated to be directly proportional to performance improvements and inversely proportional to loss — is not as straightforward as we initially hypothesized. We must recognize the intricate relationships among architectural decisions, hyperparameter configurations, and their consequent effects on model emissions and energy efficiency. Meaning, an increase in the number of shift layers does not uniformly lead to enhanced energy savings.

Another contributing factor to emission reduction is the precision of weight representation. Here, however, there is no clear trend visible in the ResNet20 configurations examined before. This suggests that this is a model-specific intricacy that needs to be tuned individually for each dataset.

These insights are corroborated when looking at the additional experiments in the appendix. Additionally to the Resnet20 on CIFAR10, we computed the Pareto fronts of our MOMF analysis on MobileNetV2 and GooGLeNet on CIFAR10. The corresponding results are shown in Figures 11 and 13. The overall Pareto optimal configurations from multiple seeds can be found in the appendix in Tables 3 and 4. Again, the shift depths are generally very low, either one or three, with two exceptions of seven and fourteen. The number of activation fraction bits is usually close to the upper bound of 32 bits.

We have also computed Pareto fronts of our MFMO approach on the Caltech101 dataset, using three architectures: ResNet20, MobileNetV2 and GooGLeNet. The results can be seen in Figures 14, 10 and 12. The configurations on the Pareto fronts are detailed in Tables 5, 6 and 7. Again, the large majority configurations have a low shift depth in the range of $s \in \{1, 2, 3, 4, 5, 6\}$. Only two

GoogLeNet configurations have a shift depth of eight and 9. This is still relatively low, given that GoogLeNet is a complex architecture with 22 layers in total.

The number of weight bits ranges from two to eight. As with the previously discussed results, this contributes to the reduction of emissions while not impacting the performance. Generally, the number of activation fraction and integer bits increases with lower shift depth and vice versa. This confirms our previous findings from the thoroughly discussed Resnet20 on CIFAR10.

## 6 CONCLUSION

In this work, we presented our Green AutoML approach towards the sustainable optimization of DSNNs through a multi-fidelity, multi-objective (MFMO) HPO framework. Our approach effectively addressed the critical intersection between advancing the capabilities of Deep Learning and environmental sustainability. By leveraging AutoML tools and integrating the environmental impact as an objective, we adeptly navigated the trade-off between model performance and efficient resource utilization.

Our experimental results focused on a better understanding of DSNNs. We successfully optimized DSNNs to achieve higher accuracy while minimizing energy consumption, surpassing the default configuration settings in both aspects. Through systematic experimentation, we identified key hyperparameters that significantly influence performance and emissions, such as shift depth and number of weight bits. By optimizing these hyperparameters, our MFMO approach did not just improve one dimension of the problem – it concurrently enhanced both model loss and energy efficiency, showcasing a balanced improvement across essential performance metrics. We have thoroughly explored the configuration space for DSNNs, introduced a Green AutoML approach for efficiency-driven model development (Tornede et al., 2023), and provided valuable insights into the design decisions impacting DSNN performance.

In this work, we conducted our experiments on the NVIDIA A100, a widely-used, state-of-the-art GPU standard in research and industry. Its computational capabilities and availability in many high-performance clusters make it a popular choice in the machine learning and deep learning communities. By using the A100, we ensure our findings are broadly applicable and relevant to real-world scenarios, aligning with hardware commonly utilized for training and deploying advanced neural networks. While testing on other hardware, such as low-power chips or alternative GPUs, could offer additional insights into hardware-specific performance trade-offs, this choice maximizes the potential impact and accessibility of our research.

While our use of CodeCarbon provides valuable insights into the emissions impact of training and inference for DSNNs, it is important to acknowledge the limitations and assumptions inherent in these measurements. CodeCarbon relies on real-time power draw metrics from tools like nvidia-smi and estimates emissions based on the regional energy mix, which we manually specified for accuracy. However, these estimates assume a steady power draw during computation and do not account for fluctuations in hardware utilization or dynamic changes in the energy grid. Taking into account these limitations, we consider CodeCarbon a reasonable approximation for measuring energy consumption. This is also reflected in recent literature, where studies have been conducted comparing CodeCarbon measurements to wattmeters that directly measure power consumption on machines (Bouza et al., 2023).

Future work will focus on revisiting our MFMO implementation to find a more efficient way for ParEGO and HyperBand to intertwine, such as by finding a more effective way to assign budgets and weights of objectives. It is a part of our efforts to lighten the computational load when computing the MFMO Pareto fronts. Further investigations will include exploring more DSNN-specific fidelity types and multi-objective algorithms to achieve even greater reductions in model emissions. We consider it especially interesting to use the number of weight bits as a fidelity type. By controlling the precision of the weight quantization, training can be sped up in the earlier fidelity while regaining as much information as possible, to use this for full training of the most promising configurations at maximum precision.

Through these future initiatives, we aim to refine our methodology and extend the environmental benefits of our optimized DSNNs, thereby contributing significantly to the sustainable advancement of AI technologies.

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

# A  APPENDIX

Table 2: Pareto optimal solutions on aggregated Pareto front of the ResNet20 architecture on CIFAR10 on three seeds, including the mean aggregated loss and emissions of the default configuration.

| Hyperpar. | Config 124 | Config 116 | Config 71 | Config 38 | Config 82 | COnfig 28 | Config 49 | Default |
|---|---|---|---|---|---|---|---|---|
| Batch Size | 127 | 127 | 127 | 128 | 127 | 128 | 128 | 128 |
| Optimizer | Ranger | Ranger | Ranger | Ranger | Ranger | Adagrad | Adagrad | SGD |
| Learning Rate | 0.0130 | 0.0327 | 0.0150 | 0.0542 | 0.0129 | 0.0548 | 0.0929 | 0.1 |
| Momentum | 0.7489 | 0.7718 | 0.1529 | 0.4983 | 0.6838 | 0.6783 | 0.4825 | 0.9 |
| Weight Decay | 0.0001 | 0.00005 | 0.0038 | 0.0001 | 0.0001 | 0.0002 | 0.0003 | 0.0001 |
| Weight Bits | 2 | 8 | 5 | 2 | 2 | 5 | 5 | 5 |
| Act. Int. Bits | 11 | 13 | 2 | 9 | 11 | 22 | 24 | 16 |
| Act. Frac. Bits | 32 | 30 | 32 | 11 | 27 | 11 | 8 | 16 |
| Shift Depth | 1 | 1 | 3 | 1 | 1 | 1 | 1 | 20 |
| Shift Type | PS | PS | Q | PS | PS | PS | PS | PS |
| Rounding | Det. | Det. | Det. | Stochastic | Det.c | Stochastic | Stochastic | Det. |
| **Loss** | 0.1127 | 0.1142 | 0.1161 | 0.1172 | 0.1176 | 0.1352 | 0.1443 | 0.3518 |
| **Emissions (e-4)** | 7.9802 | 7.9214 | 7.5967 | 7.0880 | 6.9903 | 6.8248 | 6.7444 | 7.5162 |

Table 3: Pareto optimal configurations and default DSNN instantiation of MobileNetV2 on CIFAR10

| Hyperparameter | Config 15 | Config 68 | Config 66 | Config 21 | Default |
|---|---|---|---|---|---|
| Optimizer | Adadelta | Adadelta | Adadelta | SGD | SGD |
| Learning Rate | 0.182188 | 0.186665 | 0.197817 | 0.183219 | 0.1 |
| Momentum | 0.726835 | 0.689596 | 0.1837 | 0.76337 | 0.9 |
| Weight Decay | 0.002727 | 0.003048 | 0.00306 | 0.002277 | 0.0001 |
| Weight Bits | 5 | 5 | 5 | 4 | 5 |
| Activation Integer Bits | 21 | 19 | 21 | 23 | 16 |
| Activation Fraction Bits | 31 | 31 | 32 | 16 | 16 |
| Shift Depth | 14 | 7 | 1 | 1 | 53 |
| Shift Type | PS | PS | PS | PS | PS |
| Rounding | Deterministic | Deterministic | Deterministic | Deterministic | Deterministic |
| **Loss** | 0.1683 | 0.1756 | 0.1797 | 0.9016 | 0.3017 |
| **Emissions** | 0.000755 | 0.000655 | 0.000552 | 0.000549 | 0.001656 |

Table 4: Pareto optimal configurations and default DSNN instantiation of GoogLeNet on CIFAR10

| Hyperpar. | Config 65 | Config 33 | Config 28 | Config 14 | Config 32 | Default |
|---|---|---|---|---|---|---|
| Optimizer | Adadelta | Adadelta | Ranger | Ranger | Adadelta | SGD |
| Learning Rate | 0.028997 | 0.023838 | 0.020002 | 0.058610 | 0.115941 | 0.1 |
| Momentum | 0.209328 | 0.494258 | 0.250184 | 0.673880 | 0.372339 | 0.9 |
| Weight Decay | 0.008487 | 0.006737 | 0.006691 | 0.002313 | 0.009464 | 0.0001 |
| Weight Bits | 2 | 3 | 2 | 4 | 2 | 5 |
| Activation Integer Bits | 21 | 24 | 29 | 26 | 29 | 16 |
| Activation Fraction Bits | 4 | 5 | 4 | 8 | 7 | 16 |
| Shift Depth | 1 | 1 | 1 | 1 | 1 | 22 |
| Shift Type | Q | Q | Q | PS | PS | PS |
| Rounding | Stochastic | Deterministic | Deterministic | Deterministic | Deterministic | Deterministic |
| **Loss** | 0.1347 | 0.1409 | 0.1636 | 0.1748 | 0.1850 | 0.1810 |
| **Emissions** | 0.000916 | 0.000912 | 0.000876 | 0.000840 | 0.000835 | 0.001388 |

Table 5: Pareto optimal configurations and default DSNN instantiation of ResNet20 on Caltech101

| Hyperparameter | Config 26 | Config 66 | Config 32 | Config 76 | Default |
|---|---|---|---|---|---|
| Activation Fraction Bits | 17 | 27 | 13 | 21 | 16 |
| Activation Integer Bits | 24 | 21 | 22 | 25 | 16 |
| Learning Rate (lr) | 0.023 | 0.039 | 0.076 | 0.015 | 0.1 |
| Momentum | 0.559 | 0.333 | 0.344 | 0.551 | 0.9 |
| Optimizer | Ranger | Ranger | Ranger | RMSProp | SGD |
| Rounding | Deterministic | Stochastic | Stochastic | Deterministic | Deterministic |
| Shift Depth | 2 | 2 | 1 | 1 | 20 |
| Shift Type | PS | PS | PS | Q | PS |
| Weight Bits | 2 | 5 | 2 | 5 | 5 |
| Weight Decay | 0.0033 | 0.0027 | 0.0029 | 0.0069 | 0.0001 |
| **Loss** | 0.456 | 0.532 | 0.636 | 0.874 | 0.679 |
| **Emissions** | 0.00046 | 0.00044 | 0.00044 | 0.00044 | 0.00109 |

Table 6: Pareto optimal configurations and default DSNN instantiation of MobileNetV2 on Caltech101

| Hyperparameter | Config 9 | Config 65 | Config 33 | Config 74 | Default |
|---|---|---|---|---|---|
| Activation Fraction Bits | 26 | 30 | 7 | 6 | 16 |
| Activation Integer Bits | 8 | 12 | 11 | 32 | 16 |
| Learning Rate (lr) | 0.192 | 0.199 | 0.004 | 0.006 | 0.1 |
| Momentum | 0.510 | 0.545 | 0.016 | 0.012 | 0.9 |
| Optimizer | Adadelta | Adadelta | SGD | Adam | SGD |
| Rounding | Deterministic | Deterministic | Deterministic | Stochastic | Deterministic |
| Shift Depth | 6 | 3 | 1 | 1 | 53 |
| Shift Type | PS | PS | Q | Q | PS |
| Weight Bits | 3 | 2 | 5 | 4 | 5 |
| Weight Decay | 0.009 | 0.004 | 0.004 | 0.004 | 0.0001 |
| **Loss** | 0.274 | 0.276 | 0.459 | 0.870 | 0.337 |
| **Emissions** | 0.00053 | 0.00049 | 0.00046 | 0.00046 | 0.00066 |

Table 7: Pareto optimal configurations and default DSNN instantiation of GoogLeNet on Caltech101

| Hyperpar. | Config 66 | Config 25 | Config 44 | Config 63 | Config 33 | Config 21 | Config 20 | Default |
|---|---|---|---|---|---|---|---|---|
| Activation Fraction Bits | 5 | 8 | 19 | 7 | 20 | 4 | 30 | 16 |
| Activation Integer Bits | 25 | 26 | 9 | 10 | 7 | 30 | 7 | 16 |
| Learning Rate (lr) | 0.058 | 0.059 | 0.052 | 0.13 | 0.199 | 0.187 | 0.017 | 0.1 |
| Momentum | 0.185 | 0.642 | 0.194 | 0.889 | 0.647 | 0.367 | 0.248 | 0.9 |
| Optimizer | Adadelta | Adadelta | Adadelta | Adagrad | Radam | Adagrad | Adam | SGD |
| Rounding | Det. | Det. | Det. | Det. | Det. | Stochastic | Stochastic | Det. |
| Shift Depth | 8 | 1 | 9 | 1 | 2 | 2 | 2 | 22 |
| Shift Type | Q | PS | PS | PS | PS | Q | Q | PS |
| Weight Bits | 3 | 4 | 3 | 2 | 4 | 3 | 2 | 5 |
| Weight Decay | 0.0004 | 0.0019 | 0.0029 | 0.0006 | 0.0027 | 0.0048 | 0.0059 | 0.0001 |
| **Loss** | 0.356 | 0.362 | 0.376 | 0.563 | 0.778 | 0.793 | 0.922 | 0.466 |
| **Emissions** | 0.0026 | 0.0026 | 0.0023 | 0.0022 | 0.0021 | 0.0021 | 0.0020 | 0.0027 |

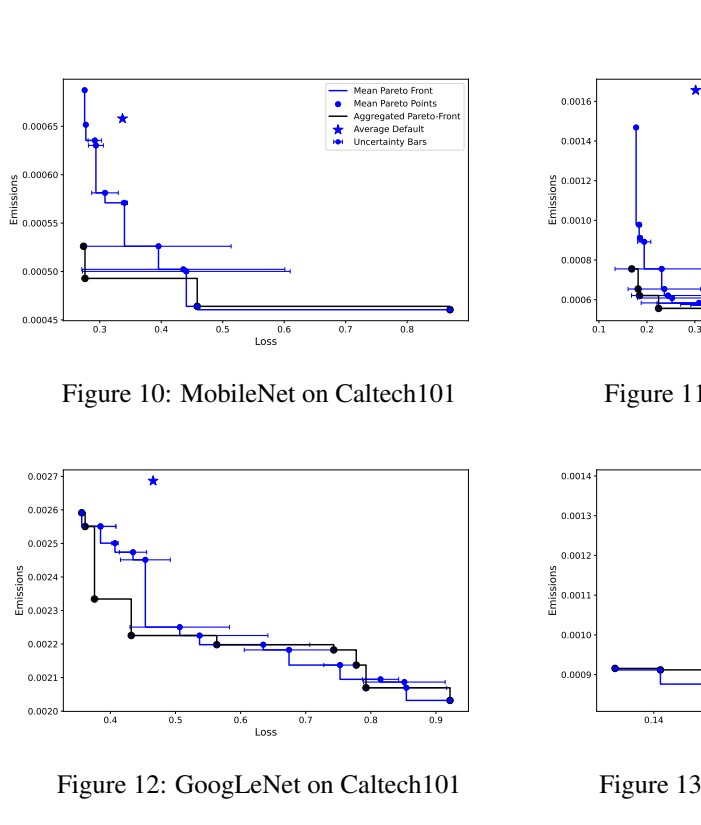

Figure 10: MobileNet on Caltech101

Figure 11: MobileNet on CIFAR10

Figure 12: GoogLeNet on Caltech101

Figure 13: GoogLeNet on CIFAR10

Figure 14: ResNet20 on Caltech101

Figure 15: ResNet20 on CIFAR10

Figure 16: Comparison of Pareto fronts for MobileNet, GoogLeNet, Resnet20 on Caltech101 and CIFAR10 datasets on multiple seed. We show the loss in % and the emissions in $gCO_2eq$. We calculate the mean Pareto front w.r.t. emissions, including error bars, as well as an aggregated Pareto front of Pareto optimal solutions from all runs.

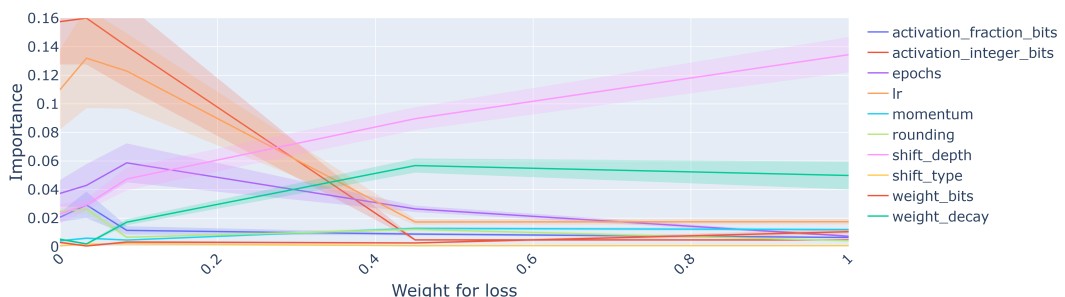

Figure 17: MO analysis of hyperparameter importance of a ResNet20 on CIFAR10 w.r.t. loss and emissions using the fANOVA method. The x-axis shows $w_l$, the weight of the objective loss, ranging from 0 to 1. The weight of the objective emissions is thus $w_e = 1 - w_l$. The y-axis shows the importance of each hyperparameter in the legend.

---

**Algorithm 1** Multi-Fidelity Optimization with Parego for DSNNs

---

**Require:** Configuration space $\mathcal{C}$, objectives $\mathcal{L} = [\mathcal{L}_{\text{loss}}, \mathcal{L}_{\text{emissions}}]$, budget range $[B_{\min}, B_{\max}]$, number of trials $N$, DSNN architecture $\mathcal{A}$.
**Ensure:** Pareto optimal configurations $\mathcal{P}_N$.
1: Initialize scenario $\mathcal{S}$ with $\mathcal{C}$, $\mathcal{L}$, $(B_{\min}, B_{\max})$, and $N$.
2: Generate initial observation dataset $\mathcal{D}_{\text{init}}$ by sampling $k$ random configurations $\{\lambda_i\}_{i=1}^{k} \subset \mathcal{C}$.
3: Define intensifier $\mathcal{H}$ as Hyperband for budget allocation.
4: Initialize optimizer $\mathcal{O}$ using Parego and $\mathcal{H}$.
5: **for** each trial $t \in \{1, \ldots, N\}$ **do**
6:     Select a configuration $\lambda_t \in \mathcal{C}$ using Parego.
7:     Allocate budget $b_t \in (B_{\min}, B_{\max})$ using $\mathcal{H}$.
8:     Perform evaluation of $\lambda_t$ with budget $b_t$:
   1. Convert DSNN $\mathcal{A}$ to a shift-based architecture $\mathcal{A}'$:

   $$\mathcal{A}' = \text{convert\_to\_shift}(\mathcal{A}, \lambda_t[\text{shift\_depth}], \lambda_t[\text{shift\_type}])$$
   $$w' = \text{round}(w, \lambda_t[\text{rounding}]), \quad \forall w \in \mathcal{A}',$$

   where convert\_to\_shift replaces standard operations with shift operations, and round applies deterministic or stochastic rounding.
   2. Train $\mathcal{A}'$ for $b_t$ epochs and compute the objective values:

   $$\mathcal{L}_{\text{loss}}(\lambda_t) = \frac{1}{|D_{\text{test}}|} \sum_{(x,y) \in D_{\text{test}}} \ell(f_{\mathcal{A}'}(x), y),$$
   $$\mathcal{L}_{\text{emissions}}(\lambda_t) = \text{measure\_emissions}(\mathcal{A}', b_t),$$

   where $D_{\text{test}}$ is the test dataset, $\ell$ is the cross-entropy loss, and measure\_emissions computes the energy consumption.
   3. Update observation dataset:

   $$\mathcal{D}_t \leftarrow \mathcal{D}_{t-1} \cup \{(\lambda_t, [\mathcal{L}_{\text{loss}}(\lambda_t), \mathcal{L}_{\text{emissions}}(\lambda_t)])\}.$$

9:     Update the Pareto front:

   $$\mathcal{P}_t = \{(\lambda, \mathcal{L}(\lambda)) \in \mathcal{D}_t \mid \nexists \lambda' \in \mathcal{D}_t : \mathcal{L}(\lambda') \succ \mathcal{L}(\lambda)\},$$

   where $\mathcal{L}(\lambda') \succ \mathcal{L}(\lambda)$ denotes that $\lambda'$ dominates $\lambda$.
10: **end for**
11: **return** $\mathcal{P}_N$

---

