# OpenReview forum: "Leveraging AutoML for Sustainable Deep Learning: A Multi-Objective HPO Approach on Deep Shift Neural Networks"
_ICLR.cc/2025/Conference — Submitted to ICLR 2025_

### Official Review · Reviewer_d92Q · 2024-11-04

**Soundness:** 3
**Presentation:** 2
**Contribution:** 2
**Rating:** 6
**Confidence:** 3

**Summary:**

The paper studies Deep Shift Neural Networks and how they can be used for computationally efficient and energy-friendly deployment. Particularly, the paper extends the prior work on DeepShiftQ and DeepShift-PS and uses a multi-fidelity, multi-objective optimization algorithm to better search the optimization space of deep shift networks while ensuring good task accuracy and energy consumption is reduced. The efficacy of the proposed algorithm is demonstrated on ResNet20 architecture.

**Strengths:**

1. The paper describes the background literature work in good details.
2. The paper discusses the details of results and importance of each choice in optimization landscape. (Section 5).

**Weaknesses:**

1.  The paper should include a section of how energy efficiency compares to tensor core operations with reduced precisions on modern GPUs (A100, H100). Since these GPUs have high specific dense tensor cores in SMs, these are very energy efficient in comparison to equivalent operations that execute on CUDA cores (this where bit shift operations would execute). If the motivation is design custom ASICS, then the paper should definitely include a section explaining this in depth.
2. The paper conducts experiments on very small number of networks that are specifically image recognition focused. The paper should definitely include more architectures specifically transformer architectures (vision transformers if the paper only wants to deal with vision related tasks).
3. Paper should include a pseudo code section describing how various components of the optimization algorithms used.
4. Figure 1 is hard to understand with few data points and huge space on x axis wasted (what is data point indicated by "x" corresponding to?).
5. The paper should include error bars in the results and use the results averaged over multiple seeds (3 seeds in this case). Since quantization methods often have higher variance, providing the mean and standard deviation of the reported numbers would provide a better understanding of the experimental results.
6. The paper seems hard to read and understand in one go for a new reader. I guess its because the core contribution requires dependence on too many methods from Background and these are not described with concrete examples in Background literature.


Nit:
The explanation of DeepShift-PS at line 146 can definitely be improved by adding an example or pseudo code.
Line 143 => tildeWq (looks like wrong text since the formula there on Line 144).

**Questions:**

Mentioned in weakness section.

---

> ### Author Response · Authors · 2024-11-21
>
> Dear reviewer,
>
> Thank you for taking the time to give detailed feedback on our paper. We appreciate your input and incorporated it into the revised version of our paper, along with the color-coded changes.
>
> Choice of Hardware (color blue)
>
> We agree that comparing the energy efficiency of our approach to tensor core operations is an important consideration. Since DSNNs rely on sparsity and reduced bit-width arithmetic, a detailed comparison would require benchmarking DSNN operations directly against tensor cores to quantify the energy efficiency gap or trade-offs. We conducted our experiments on the NVIDIA A100, a widely-used, state-of-the-art GPU. The A100 is considered a standard chip in both research and industry due to its availability and adoption across various computational settings. By focusing on this hardware, we ensure the broad applicability and relevance of our findings, as many researchers and practitioners are likely to use similar infrastructure. This choice maximizes the potential impact and accessibility of our research.
>
> Extension of Experiments to Other Architectures and Datasets (color red)
>
> We have taken your comment regarding generalizability into account. We were able to confirm our findings in additional experiments, showing the generalizability of our work. We include results from ResNet20, MobileNetV2 and GoogLeNet architectures on the Cifar10 and Caltech101 datasets on page 8, Figure 1. We show the mean pareto fronts of multiple seeds for each experiment setup. We include error bars and an aggregated pareto front showing the overall pareto-optimal configurations, as well as the averaged performance of the default configuration.
> First of all, we were able to identify configurations that provide a reduction in energy savings and loss of up to 20%, respectively. This is an increase compared to our previous results for energy efficiency. On multiple architectures across an additional dataset, we confirm our findings that it is not sufficient to simply quantize the entire architecture to obtain optimal performance and efficiency tradeoffs. The optimal configurations generally contain varying trade-offs of the DSNN-specific and other hyperparameters. This validates our assumption that we need the AutoML approach to be energy-efficient and obtain high performance with DSNNs.
>
> Pseudocode
>
> Furthermore, we incorporated your pseudo-code suggestion. Our paper now includes the pseudo-code for our AutoML approach to provide a more concise explanation. It can be found in the appendix.
>
> Graphics
>
> We adapted our graphics for our added experiments (page 8, Figure 1). We now show the mean Pareto fronts including uncertainty bars from evaluation on multiple seats. We included aggregated Pareto fronts, which include the Pareto optimal solutions from multiple seeds, as well as the averaged performance of the default configurations.
>
> Regarding the background section, we thank you for pointing out that new readers might need some time to grasp the presented concept. If you have any suggestions, which parts specifically would need more explanation, we will be happy to provide that.

---

> > ### Comment · Reviewer_d92Q · 2024-11-25
> >
> > Thanks to the authors for their detailed responses. Based on the results provided for the new tasks, I have updated my scores.

---

> > > ### Author Response · Authors · 2024-11-25
> > >
> > > Many thanks for increasing your score. Is there anything else we can do s.t. you would increase your score further?

---

### Official Review · Reviewer_t3fn · 2024-11-04

**Soundness:** 3
**Presentation:** 2
**Contribution:** 2
**Rating:** 6
**Confidence:** 3

**Summary:**

The paper aimed to address the resource and environmental challenge introduced by the great computational cost of DL models. In particular, they focused on the DNNs using shift operations. They used multi-fidelity (MF) HPO with multi-objective optimization to find some pareto-optimal designs for DSNNs. With this approach, the performance is increased by 20% with 10% less carbon emission.

**Strengths:**

AutoML based solutions can reduce the design cycle for DSNN model development.
Pareto-optimal design  is a standard way. However it offers flexibility for the performance-energy trade-off.

**Weaknesses:**

Lacks novelty. The major techniques HPO used for autoML is not new.
The improvement is marginal (only 20% for performance and 10% for energy reduction). But acceptable when compared with human experts.

**Questions:**

no

---

> ### Author Response · Authors · 2024-11-21
>
> Dear reviewer,
>
> Thank you for your comments. We would like to clarify some of our contributions.
>
> Extension of Experiments to Other Architectures and Datasets (color red in the updated paper)
>
> We were able to confirm our findings in additional experiments, showing the generalizability of our work. We include results from ResNet20, MobileNetV2 and GoogLeNet architectures on the Cifar10 and Caltech101 datasets on page 8, Figure 1. We show the mean Pareto fronts of multiple seeds for each experiment setup. We include error bars and an aggregated Pareto front showing the overall Pareto optimal configurations, as well as the averaged performance of the default configuration.
> First of all, we were able to identify configurations that provide a reduction in energy savings and loss of up to 20%, respectively. This is an increase compared to our previous results for energy efficiency. In our opinion, an increase in this order of magnitude is considerable, especially in both target dimensions. All the more when it is not unusual in comparable literature to achieve a performance increase of a few percentage points.
>
> On multiple architectures across an additional dataset, we confirm our findings that it is not sufficient to simply quantize the entire architecture to obtain optimal performance and efficiency tradeoffs. The optimal configurations generally contain varying trade-offs of the DSNN-specific and other hyperparameters. This validates our assumption that we need the AutoML approach to be energy-efficient and obtain high performance with DSNNs.
>
> We hope this response clarifies our contributions. Thank you again for your valuable feedback, and we look forward to further refining and improving our approach.

---

> > ### Author Response · Authors · 2024-11-25
> >
> > Dear reviewer,
> > The discussion phase ends soon. If there is anything else, we can do s.t. you would be willing to increase your score, please let us know.

---

### Official Review · Reviewer_h2N1 · 2024-11-07

**Soundness:** 3
**Presentation:** 2
**Contribution:** 2
**Rating:** 5
**Confidence:** 4

**Summary:**

The paper explores the optimization of Deep Shift Neural Networks (DSNNs) using Automated Machine Learning (AutoML) for balancing performance and energy efficiency. The authors propose an approach that leverages Multi-Fidelity (MF) and Multi-Objective (MO) Hyperparameter Optimization (HPO) to identify Pareto-optimal configurations of DSNNs. Their methodology integrates tools like SMAC3 and CodeCarbon to simultaneously optimize accuracy and reduce carbon emissions. The experiments conducted on architectures such as ResNet20, MobileNetV2, and GoogLeNet across datasets like CIFAR10 and Caltech10.

**Strengths:**

1. The paper offers detailed quantitative results, hyperparameter importance studies, providing a robust empirical basis for the proposed approach.

2. The motivation of the proposed method is to use shift computing to reduce the computational consumption of DSNNs. The article uses emissions as the target metric, which is interesting and has application value.

3. The work effectively showcases that AutoML can yield DSNN configurations that outperform baseline models in both performance and energy efficiency, emphasizing practical implications for low-resource settings.

**Weaknesses:**

1. For readers who are not familiar with AutoML or optimization methods, the technicality and depth of this paper may be difficult to grasp. A more detailed introduction in the Background section and a clearer writing style would help convey the motivation of the article. For instance, in Section 3, the description of DSNNs only briefly introduces the weight transformation of DSNNs through Equation 3.3 and Equation 3.4 for the simple flip(·) function. However, as introduced in [1], the core calculation of the shfit function is not explained in the article, making it difficult for readers to understand the true characteristics of DSNNs. From a writing logic perspective, there is not a strong connection between the backgrounds in Sections 3.1-3.3, and there are several typos in the article, such as '$tildeW_q$' in line 143.

2. The core contribution of the paper is the ability to quantify the emissions impact of training and inference of DSNNs, a topic that is environmentally friendly. However, energy efficiency is dependent on the chosen computing platform. The authors conducted tests using the A100. The reliance on energy and emissions tracking tools like CodeCarbon is beneficial, but the paper could improve by discussing limitations or assumptions inherent in these measurements and potential variability due to different hardware.

 3. A small-size model like ResNet20 on A100 may not be comprehensive enough, and researchers would like to see the relationship between the computational energy of different types of neural network structures on various energy-efficient computing devices and the shift configuration.

[1] DeepShift: Towards multiplicationless neural networks.

**Questions:**

I am curious about how the authors deployed shift-based computations instead of traditional addition and multiplication on the NVIDIA A100 GPU. Can this type of computation be extended to more networks, even though LLMs may be larger in size, what about smaller BERT models?

---

> ### Author Response · Authors · 2024-11-21
>
> Dear reviewer,
>
> Thank you for your insightful comments regarding the writing style. We appreciate your feedback and have taken it into account to improve the clarity and structure of our paper. In particular, we have refined the \textcolor{cyan}{background section} to ensure it provides a clearer and more comprehensive foundation for our work. We hope these revisions enhance the readability and overall presentation of the paper. We color-coded some changes in our updated paper, matching our corresponding answers.
>
> Extension of Experiments to Other Architectures and Datasets (color red)
>
> We agree with your comment that researchers would like to see the results of our approach on multiple architectures. We provide this in the revised version of our paper. We were able to confirm our findings in additional experiments, showing the generalizability of our work. We include results from ResNet20, MobileNetV2 and GoogLeNet architectures on the Cifar10 and Caltech101 datasets on page 8, Figure 1. Tables listing the Pareto optimal configuration for each architecture on either dataset can be found in the appendix. We show the mean pareto fronts of multiple seeds for each experiment setup. We include error bars and an aggregated pareto front showing the overall pareto-optimal configurations, as well as the averaged performance of the default configuration.
> We were able to identify configurations that provide a reduction in energy savings and loss of up to 20%, respectively. This is an increase compared to our previous results for energy efficiency. On multiple architectures across an additional dataset, we confirm our findings that it is not sufficient to simply quantize the entire architecture to obtain optimal performance and efficiency.
>
> Measurements of Energy Consumption (color purple)
>
> While our use of CodeCarbon provides valuable insights into the emissions impact of training and inference for DSNNs, it is important to acknowledge the limitations and assumptions inherent in these measurements. CodeCarbon relies on real-time power draw metrics from tools like nvidia-smi and estimates emissions based on the regional energy mix, which we manually specified for accuracy. However, these estimates assume a steady power draw during computation and do not account for fluctuations in hardware utilization or dynamic changes in the energy grid.
> Taking into account these limitations, we consider CodeCarbon a reasonable approximation for measuring energy consumption. This is also reflected in recent literature, where studies have been conducted comparing CodeCarbon measurements to wattmeters that directly measure power consumption on machines (Bouza et al., 2023).
>
> Choice of Hardware (color blue)
>
> We conducted our experiments on the NVIDIA A100, a widely-used, state-of-the-art GPU. The A100 is considered a standard chip in both research and industry due to its availability and adoption across various computational settings. By focusing on this hardware, we ensure the broad applicability and relevance of our findings, as many researchers and practitioners are likely to use similar infrastructure. This choice maximizes the potential impact and accessibility of our research.

---

> > ### Author Response · Authors · 2024-11-25
> >
> > Dear reviewer,
> > The discussion phase ends soon. If there is anything else, we can do s.t. you would be willing to increase your score, please let us know.

---

> > ### Comment · Reviewer_h2N1 · 2024-11-26
> > **keep my score**
> >
> > I appreciate the efforts the author has put into revising the article, but in the revised version, the author still does not provide a detailed computation process of the shift function in matrix operations in DSNNs. Only citing the article is insufficient; in reality, this is much more important than Equation 3.1 and Equation 3.2.
> >
> > Regarding my question about extending experiments to more architectures, I am particularly interested in knowing if this computational method is equally efficient in BERT or Transformer calculations since the core components of LLM are based on these structures, and addressing energy efficiency in LLM is more critical compared to traditional CNNs. To save the author's time, I hoped to see experiments on small-scale Transformer or BERT models (but not LLM), which I did not come across. Additionally, both in industry and academia, there is limited research on the energy consumption of small-scale models like ResNet on A100 GPUs. Conversely, the application of these models is more significant in consumer-grade GPUs or edge scenarios.
> >
> > I will maintain my score and hope that this work undergoes more profound revisions. I believe that with improvements in the experimental setup and writing quality, this work has the potential to be promising.

---

> ### Author Response · Authors · 2024-11-27
>
> Dear reviewer,
>
> we agree that the computation of the shift operations in DSNNs is crucial for understanding our approach. However, can you please clarify which part of the computation from the original paper you are referring to exactly? We described the matrix quantization in Equations 3.1 and 3.2 for both the DeepShift-Q and DeepShift-PS approach. Are you referring to the gradient computations during forward and backward pass? We are happy to improve the paper along the lines of your recommendations. Thank you in advance for the clarification.
>
> Regarding the relevance of our approach for different architectures, we would like to emphasize the relevance of energy efficiency for training CNNs from scratch, which are still widely used in practice, in particular in embedded or edge devices that do not always allow running large-scale transformer models or to communicate with the cloud.

---

### Official Review · Reviewer_JrHz · 2024-11-08

**Soundness:** 3
**Presentation:** 3
**Contribution:** 2
**Rating:** 5
**Confidence:** 3

**Summary:**

The paper reports on the tuning and optimization of deep shift neural networks (DSNNs) for both accuracy and energy efficiency. Automatic hyperparameter and architecture search is employed to tune models and explore the Pareto front of the accuracy/energy tradeoff for the models. The paper defines the relevant search spaces and discovers configurations improving upon the state-of-the-art for DSNNs.

**Strengths:**

1. The paper is clear and well-written and provides both sufficient background and context for the work.
2. The paper is targeting improved efficiency for DSNNs, which has the potential for positive environmental impact.
3. The paper identifies improved configurations for DSNNs, and explores the Pareto front to understand the tradeoffs between configurations.
4. The impact of specific hyperparameters are analyzed.

**Weaknesses:**

1. It is not clear to me what the novelty and broader impact of the work is. It seems, essentially, to use relatively standard automatic tuning approaches to optimize an existing architecture. While this provides some value, it also seems very specific to the DSNNs considered here. To what extent are there generalizable lessons or takeaways from the work that are more broadly applicable?
2. I am unclear on how exactly energy efficiency is measured in order to incorporate it into the optimization process, which is critical for evaluating the paper's methodology (and might be useful for others). CodeCarbon typically requires assumptions about both the hardware being used and the regional energy mixture, as well as experimental measurements. The paper should provide some additional details and discussion on this.
3. Some of the hyperparameter importance measurements w.r.t. emissions (Figure 3) seem strange to me. In particular, how does the learning rate impact the emissions?

**Questions:**

Please see the comments/questions above under "Weaknesses".

---

> ### Author Response · Authors · 2024-11-21
>
> Dear reviewer,
>
> We appreciate your comments on our paper and the opportunity to improve it. Thank you for acknowledging the clarity and writing style of our paper. We would like to clarify our findings regarding your feedback. We updated our paper and color-coded some changes to make it easier for you to match them to our answers and your questions.
>
> Generalization to Other Quantization Architectures (color red)
>
> We currently focus on DSNNs because they are promising quantization architectures for energy efficiency that can be leveraged in various use cases. It is very versatile since it can be used to convert the backbone architecture of numerous pipelines and tasks. The rounding of weights and activations to their respective bit representations, and the reductions of its precision, is a relatively standard approach in quantization techniques. This foundational similarity makes our findings applicable beyond DSNNs to other quantization-based architectures. By focusing on the intricate optimization of quantization parameters—such as bit precision and shift depth—our approach provides generalizable insights that can enhance the efficiency and performance of a wide range of quantized architectures. Thus, while DSNNs serve as our primary focus, the principles and methodologies developed in this work have broader applicability to other quantization approaches.
>
> Measurements of Energy Consumption (color purple)
>
> To measure energy consumption in our experiments, we used CodeCarbon in conjunction with SLURM, the workload manager on our HPC cluster. CodeCarbon was integrated into the SLURM job script to monitor energy usage throughout the task's execution. For energy monitoring, CodeCarbon leverages NVIDIA’s Nvidia-smi to capture the real-time power draw of the GPUs on the allocated node. It calculated total energy consumption by multiplying the power draw by the job’s runtime, which was determined based on the start and end times recorded during execution. To estimate carbon emissions, we manually specified the regional energy mix, ensuring accurate carbon intensity values were used. These values were combined with the calculated energy consumption to determine the emissions generated by the job. This approach allowed us to focus specifically on GPU energy usage during training while accounting for the cluster's local energy profile. We will add details about our local energy profile in later revisions of the paper so as not to reveal parts of our identity. However, we believe that we can say that our assumed local energy profile is fairly typical for industry nations these days.
>
> Lastly, we would like to address the hyperparameter importances, particularly the learning rate. Looking back at our experiments, we suspect that a poorly tuned learning rate can lead to suboptimal weight distributions, which may increase the computational cost of inference. For instance, inefficient weight distributions may require more operations or memory accesses during inference, indirectly raising emissions. Conversely, an optimal learning rate facilitates faster convergence to well-optimized weights, reducing computational overhead and energy usage during inference.

---

> > ### Author Response · Authors · 2024-11-25
> >
> > Dear reviewer,
> > The discussion phase ends soon. If there is anything else, we can do s.t. you would be willing to increase your score, please let us know.

---

> > ### Comment · Reviewer_JrHz · 2024-11-26
> >
> > Thank you for the response.
> >
> > Generalization: Can you be more precise about what the generalizable insights are? It seems very clear that, for any application of quantization, one should carefully select the bitwidth, but it is unclear, to me at least, what lessons to take away from this paper.
> >
> > Energy measurements: Thanks for the clarification, this helps some. Is it really realistic to use a local energy mixture for operations that would run in a data center? These typically have very different energy mixes. Further, how is energy efficiency during inference incorporated? Especially in the case where inference may be on very different devices than during training, and may be performed in areas with different energy mixes.
> >
> > Hyperparameter importance: Thanks, this makes some intuitive sense; however, it would seem worthwhile to verify this experimentally.
> >
> > Overall while this work is promising, I do not think it's ready yet and am maintaining my current score.

---

> > > ### Author Response · Authors · 2024-11-29
> > >
> > > Dear reviewer,
> > >
> > > We appreciate your questions and address them in the following:
> > >
> > > Generalizability: The generalizable insights from our work extend beyond the need to carefully select bitwidths for quantization. Our results reveal that optimal configurations for DSNNs are often counterintuitive and highly dependent on the intricate relations between hyperparameters. For example, we found that low shift depths often achieve superior trade-offs between accuracy and energy efficiency, challenging assumptions about full quantization of networks. Additionally, our analysis highlights the importance of prioritizing specific hyperparameters for different objectives, providing a targeted approach to DSNN optimization. These findings are consistent across multiple architectures and datasets, demonstrating their broader applicability. Overall, our study offers a systematic, automated approach to detecting these insights, which are relevant not only for DSNNs but also for other quantized and resource-efficient neural networks.
> > >
> > > Energy Measurements: We acknowledge that energy consumption varies significantly based on regional energy mixes and the infrastructure of data centers. However, our approach efficiently allows for the recalculation of Pareto fronts for specific locations or hardware setups with different energy profiles. This adaptability ensures that stakeholders can derive actionable insights under varying conditions.
> > > While data centers may often rely on carbon-neutral energy sources, the broader applicability of our method extends to embedded devices, such as those used in automobiles or IoT systems. These devices operate in highly varied environments, with energy sources dependent on local conditions. Our approach enables stakeholders to determine location-specific or even time-sensitive Pareto fronts, providing a tailored basis for decision-making. This empowers users to balance trade-offs between accuracy and emissions based on financial, ecological, or operational considerations.
> > >
> > > Emissions are typically calculated by multiplying the energy consumed by a carbon intensity factor.
> > > For a given energy mix, this factor remains constant. As a result, the emissions scale linearly with energy consumption when the energy mix is fixed. Thus, once we compute a Pareto front based on energy consumption, the emission axis can be scaled accordingly to reflect the specific carbon intensity of the region.
> > > In dynamic scenarios where energy mixes change over time, our approach enables rapid recalculation of Pareto fronts to reflect these variations so that stakeholders can accommodate real-time or regional changes in energy conditions. This ensures our method remains practical and relevant in diverse and evolving settings.
> > >
> > > HP Importance: Following your comment, we verified our assumption about learning rates affecting the emissions experimentally. We instantiated a ResNet20 with the default configuration and 95 learning rates, randomly sampled from the original search space. An analysis containing further uncertainty estimates will be included in the camera-ready version.
> > >
> > > Analyzing the resulting emission leads to a mean value of 0.0009873 gCo2eq, which is slightly larger than the range of our Pareto fronts. 82 of those values are below 0.0008, which is the supremum of our Pareto fronts. However, while the majority of the resulting emissions are in the expected range, 13 emission values are significant outliers in a boxplot analysis, causing a standard deviation of 0.0009571 gCo2eq. This leads to the conclusion that the learning rate is indeed an influential HP on the emissions data since it can drastically lead to an increase in emissions when chosen suboptimally. Our AutoML approach offers a solution to exactly this problem, since we aim to automate both the identification and optimal choice of hyperparameters.

---

### Official Review · Reviewer_Ak52 · 2024-11-08

**Soundness:** 1
**Presentation:** 2
**Contribution:** 1
**Rating:** 5
**Confidence:** 5

**Summary:**

This paper investigates the potential of DSNNs to reduce computational complexity and environmental impact in deep learning (DL) models. Through AutoML techniques, the authors explore ways to optimize DSNNs by balancing multiple objectives, including accuracy and energy consumption. By applying multi-fidelity hyperparameter optimization and multi-objective optimization, they identify Pareto-optimal configurations that improve DSNN performance by 20% while reducing emissions by 10% compared to default models. Notably, their findings challenge traditional expectations around quantization, showing that selectively quantizing smaller portions of the network at low precision can yield the best trade-offs between accuracy and energy savings. These results, validated across different model architectures, underscore DSNNs' potential in designing energy-efficient, high-performing DL models.

**Strengths:**

- The paper sheds light on an interesting design for quantized neural networks, DSNNs, which significantly reduce the computational demands at inference, positioning them as a sustainable AI solution.
- Through multi-fidelity hyperparameter optimization and multi-objective optimization, the proposed approach achieves configurations that improve DSNN performance by 20% while reducing emissions by 10% over baselines.
- The findings challenge conventional beliefs in model quantization, showing that selectively quantizing smaller portions of the network at low precision yields optimal trade-offs between energy efficiency and model accuracy.

**Weaknesses:**

- The paper's contribution in terms of novelty is somewhat limited and could benefit from further clarification. The proposed optimization approach for DSNNs appears to directly apply existing optimization techniques (DGEMO and SMAC) for hyperparameter tuning, without substantial adaptation or refinement specific to the DSNN topology or its unique optimization landscape. As it stands, the work reads more as a straightforward application of existing AutoML methods rather than an innovative extension or advancement in the field. To improve the paper's originality, I suggest the authors consider developing an algorithm that more effectively integrates various design dimensions, such as backbone architecture and bit-shifting hyperparameters, with specific tailoring for the DSNN architecture. This could provide a more distinct and impactful contribution to the AutoML research.
- The paper would benefit from a more detailed architectural analysis, particularly w.r.t. the types of layers being quantized and the specific optimizations for bit-shifting. Currently, there is a lack of discussion on how bit-shift quantization affects various layers, which limits the depth of the architectural insights provided. To strengthen the contribution, I suggest the authors include a breakdown of the impact of bit-shift quantization on different layers and specify precisely which layers are being quantized.
- The evaluation section is somewhat limited, as it currently includes results on only a single model architecture (ResNet20). This narrow focus makes it challenging to draw robust conclusions about the effectiveness and generalizability of the proposed AutoML framework. To strengthen the evaluation, I recommend that the authors test the approach on a broader range of models and datasets. This would provide a more comprehensive assessment of its applicability and impact across different scenarios.
- The discussion on hyperparameter importance is valuable; however, its placement later in the paper limits its impact. Presenting this ablation study earlier would help justify the choice of hyperparameters targeted for optimization, especially since only a few key design hyperparameters significantly influence the results while others contribute minimally. Moving this section forward and providing either empirical or theoretical justification for selecting these specific hyperparameters would enhance the clarity and rigor of the paper's contribution.

**Questions:**

- Could you clarify the novelty of your optimization approach for DSNNs? Specifically, are there any unique adaptations or modifications in the way DGEMO and SMAC are applied to DSNNs, beyond existing AutoML methods?
- Can you provide more details on the types of layers being quantized and the specific impact of bit-shift quantization across these layers? Are certain layers more affected by quantization than others?
- Have you considered testing your framework on additional model architectures and datasets to demonstrate its generalizability? If not, could you share insights into why ResNet20 was chosen as the sole model for evaluation?
- Could you elaborate on the rationale for the specific hyperparameters chosen for optimization? Given that only a few design hyperparameters appear to have a significant effect on the results, did you consider omitting those with minimal contributions?

---

> ### Author Response · Authors · 2024-11-21
>
> Dear reviewer,
>
> Thank you for your insightful comments. We appreciate your feedback and incorporated it into our paper to the best of our ability in this limited timeframe. We have color-coded our changes in our paper to make it easier for you to find the sections corresponding to your questions and our answers.
>
> Extension of Experiments to Other Architectures and Datasets (color red)
>
>
> First of all, we extended our evaluation of our approach. We include results from ResNet20, MobileNetV2, and GoogLeNet architectures on the Cifar10 and Caltech101 datasets on page 8, Figure 1. We show the mean Pareto fronts of multiple seeds for each experiment setup. We include error bars and an aggregated Pareto front showing the overall Pareto optimal configurations, as well as the averaged performance of the default configuration.
>  First of all, we were able to identify configurations that provide a reduction in energy savings and loss of up to 20%, respectively. This is an increase compared to our previous results for energy efficiency. On multiple architectures across an additional dataset, we confirm our findings that it is not sufficient to simply quantize the entire architecture to obtain optimal performance and efficiency tradeoffs. The optimal configurations generally contain varying trade-offs of the DSNN-specific and other hyperparameters. This validates our assumption that we need the AutoML approach to be energy-efficient and obtain high performance with DSNNs. We showed this finding of ours to be generalizable across several architectures and datasets.
>
> Implications of Quantization on the Architecture
>
> Furthermore, we thank you for suggesting further investigation into which types of layers are being quantized and what the implications across a chosen architecture are. While this is not extractable from our current setup, we have started looking into this as an extension to our current work. We believe it is valuable to look at this from a neural architecture search point of view. In future work, finding a systematic way of quantizing specific blocks of a neural architecture or using quantization for a specific neural network might enable us to improve efficiency and performance further.
>
> Hyperparameter Importance (color magenta)
>
> We included the listed hyperparameters in our configuration space to cover generic and DSNN-specific hyperparameters. We analyzed which were more relevant for the optimization process regarding performance and efficiency. We suspected that different hyperparameters influence either objective to a different extent. To confirm this, we performed HPO on the entire configuration space. Hence, we did not omit the less significant hyperparameters from our results. However, with the analysis of hyperparameter importance in our study, we offer a baseline of which hyperparameters to include for future training and inference purposes. Including only the significant ones is a promising way of further boosting the energy efficiency of the DSNNs and the optimization process (cp. ‘Tunability: Importance of Hyperparameters of Machine Learning Algorithms‘, Probst et al., 2019).
>
> MOMF Approach
>
> The novelty in our paper lies rather in the insights we gained about quantized architectures and how to navigate trade-offs between performance and efficiency. These differ from the usual intuition of quantizing an entire network for maximum efficiency. We integrated the MO and MF functionalities in SMAC since we are not aware of any off-the-shelf tool to perform MOMF more efficiently; nevertheless, our results do not depend on this optimizer choice, and thus we have emphasized this part heavily.
>
> By incorporating your remarks, we hope we have sufficiently answered your questions. We will be happy to address additional feedback and questions. We appreciate this opportunity to improve our work and hope we did so to your satisfaction.

---

> > ### Author Response · Authors · 2024-11-25
> >
> > Dear reviewer,
> > The discussion phase ends soon. If there is anything else, we can do s.t. you would be willing to increase your score, please let us know.

---

> ### Comment · Reviewer_Ak52 · 2024-11-25
>
> Thank you to the authors for providing detailed clarifications. While I appreciate the additional insights, I still have concerns regarding the paper's novelty. The findings still rely on empirical evaluations, which are influenced by the experimental setup, optimizer parameters, and architectural configurations. Furthermore, the theoretical aspects of architecture optimization should be discussed in greater depth to offer meaningful insights into DSNN. I appreciate the effort the authors have made to address my initial comments and have adjusted my score accordingly. However, I believe the paper would still benefit from further refinements and improvements.

---

> > ### Author Response · Authors · 2024-11-25
> >
> > Dear Reviewer,
> >
> > Thank you very much for improving your score.
> > (Maybe not surprisingly,) we believe that our paper is a valuable contribution to showing how much, and under surprising conditions, DSNNs can consistently perform much better than previously believed. Even with the deep theoretical knowledge of the original authors, our approach was able to outperform the original configuration of DSNNs on several benchmarks clearly. We agree that there seems to be something theoretical to discover here based on our results; however, we also believe that our paper is the important first step in this direction, which can spark further important (maybe theoretical) future work.
> > Please let us know if there is anything we can do to convince you at the last minute to increase your score further.

---

### Meta-Review · Area_Chair_rSKe · 2024-12-20

**Metareview:**

This paper presents an exploration into optimizing Deep Shift Neural Networks (DSNNs) for balancing performance and energy efficiency using Automated Machine Learning (AutoML) techniques. The authors propose a multi-fidelity, multi-objective hyperparameter optimization approach to identify Pareto-optimal configurations for DSNNs, aiming to reduce computational complexity and environmental impact.

One of the primary concerns raised by the reviewers is the lack of novelty in the optimization techniques applied. While the paper claims to offer flexibility for the performance-energy trade-off, the major techniques used for AutoML are not new, and the improvement in performance and energy reduction, though present, is considered marginal. The paper's contribution in terms of novelty is somewhat limited and could benefit from further clarification. The proposed optimization approach for DSNNs appears to directly apply existing optimization techniques without substantial adaptation or refinement specific to the DSNN topology or its unique optimization landscape. The comparison with existing methods and the generalizability of the findings were also noted as insufficient by some reviewers. The evaluation section is limited, as it currently includes results on only a few model architectures, which makes it challenging to draw robust conclusions about the effectiveness and generalizability of the proposed AutoML framework. The paper would benefit from testing the approach on a broader range of models and datasets to provide a more comprehensive assessment of its applicability and impact across different scenarios.

The reviewers expressed mixed opinions, with some leaning towards acceptance due to the relevance and potential of the work in the field of sustainable deep learning, while others were more skeptical due to the lack of innovation and concerns about the practicality of the proposed methods. The paper shows promise, but it would benefit from further refinement, particularly in addressing the concerns about hardware implementation, providing more robust comparisons with existing methods, and demonstrating the generalizability of the approach across different architectures and datasets.

This paper received an averaged score of 5.4, which is not competitive among this year's submissions. Given the balance of strengths and weaknesses, the final recommendation is to reject this submission in its current form. The paper has the potential to make a significant contribution to the field of sustainable deep learning, but it would benefit from an additional round of revisions that address the concerns about the novelty of the approach, the generalizability of the findings, and the robustness of the comparisons with state-of-the-art models.

**Additional Comments On Reviewer Discussion:**

During the rebuttal period, the primary points of discussion among reviewers centered on the novelty and generalizability of the paper's approach to optimizing Deep Shift Neural Networks (DSNNs) using Automated Machine Learning (AutoML). Reviewers were concerned that the techniques employed were not sufficiently novel, as they appeared to be direct applications of existing optimization methods without significant adaptation for DSNNs. In response, the authors attempted to clarify the unique contributions of their work, emphasizing the counterintuitive findings regarding the optimization of DSNNs and the importance of specific hyperparameters in achieving performance and efficiency trade-offs.

Another significant point of contention was the limited evaluation of the proposed method on a narrow set of architectures and datasets. The authors addressed this by extending their experiments to include additional architectures and datasets, demonstrating the generalizability of their findings across different models. While this additional evidence was noted, some reviewers remained unconvinced of the broad applicability of the results due to the still relatively limited scope of the evaluation.

In weighing these points, the final decision took into account the authors' efforts to address the reviewers' concerns, but also the persisting doubts about the paper's innovation and the breadth of its findings. The paper's contribution was deemed promising but not sufficiently robust to merit acceptance in its current form, leading to the recommendation for rejection with the suggestion for further refinement and expansion of the experimental evaluation.

---

### Decision · Program_Chairs · 2025-01-22

Reject